# Self-Tumor Antigens in Solid Tumors Turned into Vaccines by α-gal Micelle Immunotherapy

**DOI:** 10.3390/pharmaceutics16101263

**Published:** 2024-09-27

**Authors:** Uri Galili

**Affiliations:** Department of Medicine, Rush University Medical Center, Chicago, IL 60612, USA; uri.galili@rcn.com; Tel.: +1-312-753-5997

**Keywords:** cancer immunotherapy, anti-Gal antibody, α-gal epitope, α-gal micelles, α-gal glycolipids, AGI-134, APC recruitment, α-gal therapy, tumor antigens

## Abstract

A major reason for the failure of the immune system to detect tumor antigens (TAs) is the insufficient uptake, processing, and presentation of TAs by antigen-presenting cells (APCs). The immunogenicity of TAs in the individual patient can be markedly increased by the in situ targeting of tumor cells for robust uptake by APCs, without the need to identify and characterize the TAs. This is feasible by the intra-tumoral injection of α-gal micelles comprised of glycolipids presenting the carbohydrate-antigen “α-gal epitope” (Galα1-3Galβ1-4GlcNAc-R). Humans produce a natural antibody called “anti-Gal” (constituting ~1% of immunoglobulins), which binds to α-gal epitopes. Tumor-injected α-gal micelles spontaneously insert into tumor cell membranes, so that multiple α-gal epitopes are presented on tumor cells. Anti-Gal binding to these epitopes activates the complement system, resulting in the killing of tumor cells, and the recruitment of multiple APCs (dendritic cells and macrophages) into treated tumors by the chemotactic complement cleavage peptides C5a and C3a. In this process of converting the treated tumor into a personalized TA vaccine, the recruited APC phagocytose anti-Gal opsonized tumor cells and cell membranes, process the internalized TAs and transport them to regional lymph-nodes. TA peptides presented on APCs activate TA-specific T cells to proliferate and destroy the metastatic tumor cells presenting the TAs. Studies in anti-Gal-producing mice demonstrated the induction of effective protection against distant metastases of the highly tumorigenic B16 melanoma following injection of natural and synthetic α-gal micelles into primary tumors. This treatment was further found to synergize with checkpoint inhibitor therapy by the anti-PD1 antibody. Phase-1 clinical trials indicated that α-gal micelle immunotherapy is safe and can induce the infiltration of CD4+ and CD8+ T cells into untreated distant metastases. It is suggested that, in addition to converting treated metastases into an autologous TA vaccine, this treatment should be considered as a neoadjuvant therapy, administering α-gal micelles into primary tumors immediately following their detection. Such an immunotherapy will convert tumors into a personalized anti-TA vaccine for the period prior to their resection.

## 1. Introduction

Self-tumor antigens (TAs) can serve as the “Achilles heel” of malignancies in cancer immunotherapy. The TAs in this review are known and unknown antigens presented uniquely on tumor cells, including products of mutated genes, products of oncogenic viruses, aberrantly expressed cellular proteins, altered glycoproteins and glycolipids, and oncofetal antigens. The induction of protective T cell and B cell responses against TAs may result in successful immunotherapy [1,2,3,4,5]. This notion has been supported by observations in a variety of cancers, including colon carcinoma [6,7], ovarian carcinoma [8] and breast carcinoma [9,10]. Retrospective studies in patients with these types of cancer indicated that the patients had much longer survival times and an absence of metastases if their resected tumors displayed an infiltration of many T cells, in comparison to patients displaying low or no infiltration of T cells into the tested tumors. The high infiltration of T cells in tumors of patients with good prognosis was specific to the malignant tissues and not found in normal tissues. This implied that the protective immune response associated with the lack of metastases was the result of a specific immune response to TAs present on malignant cells and absent from normal cells. In contrast, the poor prognosis observed in patients with low or no T cell infiltration into tumors (also called “cold tumors”) suggested that, in these patients, the immune system failed to react effectively against the patient’s TAs and thus prevent the formation of metastases and the progression of the disease. 

The activation of T cells against tumor cells presenting TAs requires the intra-tumoral uptake of TAs by antigen-presenting cells (APC) such as dendritic cells and macrophages. These APCs process and present TA peptides on HLA molecules and transport the presented TA peptides to regional lymph nodes for the activation of TA-specific T cells. The absence of infiltrating T cells in cold tumors may be the result of various factors, including an immunosuppressive microenvironment within the lesion, the prevention of the infiltration of APCs into the tumor, the poor presentation of TAs by the tumor cells, and immune tolerance to TAs [11,12]. It has been further suggested that most cancer vaccines have not shown significant clinical benefit because of poor uptake of vaccinating TAs by APCs [12]. 

The accumulated experience in cancer immunotherapy led to the realization that a large proportion of the TAs are formed by multiple mutations specific to the individual patient. These mutations are generated due to genomic instability in tumor cells [4,12,13,14]. Many of these mutations alter amino acids within various proteins and convert them into TAs that may be targets for immunotherapy. The combination of “next generation sequencing” in tumors from individual cancer patients and algorithms developed in recent years have enabled the identification of immunogenic TA peptides (i.e., peptides with a high affinity to HLA molecules for effective presentation) that may be synthesized as “neoantigens” for the use as personalized vaccine in cancer patients [15,16,17,18]. It remains to determine the efficacy of this novel method in cancer immunotherapy. 

The present review describes an alternative simple method for inducing a personalized protective immune response against all types of TAs in the individual cancer patient. This method, which uses the self-tumor as the source of immunizing TAs, achieves the following objectives of cancer vaccines, including: (1) The use of all self-TAs as a vaccine without the need for identifying them; (2) effective targeting of the immunizing TAs for robust uptake by APCs; (3) the presentation by APC of neoantigen peptides with high affinity to HLA; (4) overcoming the immune tolerance to self-TAs. These vaccines are generated by injecting α-gal micelles into solid tumors. The α-gal glycolipids comprising these micelles spontaneously insert into the cell membrane of tumor cells in treated lesions. These glycolipids bind the abundant natural anti-Gal antibody [19], which targets tumor cells and cell membranes expressing the full range of TAs for robust uptake by APCs and their transport to regional lymph nodes. The processing and presentation of TA peptides by these APCs within lymph nodes induces a protective immune response against metastatic tumor cells. 

As detailed in this review, the main advantages of this vaccination immunotherapy with α-gal micelles are as follows: (1) the simplicity of the treatment, which involves only the intra-tumoral injection of α-gal micelles and does not require any in vitro culturing of dendritic cells and pulsing these cells with TAs or fusing them with tumor cells; (2) the possible use of this vaccine as both a neoadjuvant immunotherapy and for inducing a protective immune response against detectable metastates; (3) the ability to elicit an immune response against the full range of TAs in each patient (i.e., a personalized vaccine) without the need to identify any of the TAs; and (4) the ability of this therapy to synergize with checkpoint-inhibitor therapies. 

## 2. The Natural Anti-Gal Antibody and the α-gal Epitope

Anti-Gal is an antibody that is naturally produced throughout life in humans [19,20] in response to antigenic stimulation by gastrointestinal bacteria [21,22,23]. Anti-Gal is produced in all humans who are not severely immunocompromised. It constitutes ~1% of serum immunoglobulins [20] and as many as 1% of B cells can produce this antibody [24]. In fetal and newborn blood, anti-Gal is found as maternal IgG, whereas in children and adults, it is found as IgG, IgM, and IgA classes [20,25,26,27,28]. Anti-Gal is a polyclonal antibody [20], and its ligand is a carbohydrate antigen called the “α-gal epitope” with the structure Galα1-3Galβ1-4GlcNAc-R on mammalian glycolipids, glycoproteins and proteoglycans [29,30,31,32]. In blood-type A and O individuals, anti-Gal also comprises > 85% of anti-blood group B antibody activity [30]. The α-gal epitope is naturally expressed on the cell-surface carbohydrate chains (glycans) of nonprimate mammals, prosimians (lemurs), and New World monkeys, in which it is synthesized by the glycosylation enzyme α1,3galactosyltransferase (α1,3GT) [33,34]. The *GGTA1* gene encoding α1,3GT is found only in mammals [35,36,37] and is absent in birds, reptiles, amphibians and fish; therefore, non-mammalian vertebrates lack α-gal epitopes. The *GGTA1* gene was inactivated in ancestral Old World primates 20–30 million years ago [36,37,38]; therefore, humans, apes, and Old World monkeys lack the ability to synthesize α-gal epitopes, but they produce the natural anti-Gal antibody [39].

Anti-Gal is a potent antibody in several immune processes. Anti-Gal binding in vitro to cells presenting α-gal epitopes can mediate the killing of the cells both by complement dependent cytolysis (CDC) and by antibody-dependent cell cytolysis (ADCC) [40,41,42]. The potential of this antibody to mediate in vivo CDC and ADCC has been demonstrated in xenotransplantation studies in which pig organs are transplanted into humans or monkeys. The binding of the recipients’ anti-Gal to α-gal epitopes on pig cells in the re-perfused organs leads to rapid (hyperacute) rejection due to cell destruction by these two mechanisms [40,43,44,45]. The in vivo anti-Gal/α-gal epitope interaction provides the possibility of harnessing the immunologic potential of this antibody in a number of α-gal-associated therapies in several clinical settings, including the accelerated scar-free regeneration of injuries in the skin [46,47,48], the regeneration of injured ischemic myocardium following myocardial infarction [49,50], the regeneration of injured spinal cord [50,51] and the conversion of self-TAs into anti-tumor vaccines, which are discussed in the present review.

## 3. Anti-Gal-Mediated Targeting of Tumor Cell Vaccines Presenting TAs and α-gal Epitopes to APCs

As mentioned above, a major cause for the lack of an effective immune response to TAs in cancer patients is the inability of APCs to identify tumor cells presenting TAs as cells or cell membranes that “ought” to be internalized for the processing and presentation of TA peptides [11]. Similar to any other vaccine, TA vaccines must be internalized by APCs in order to process the vaccinating antigens and present the TA peptides on the HLA molecules of these APCs. The APCs further transport the presented peptides to regional lymph nodes, where they activate TA specific CD8+ T cells which proliferate, leave the lymph node, circulate in the body and kill metastatic TA-presenting tumor cells. In addition, these APCs activate CD4+ T cells that provide help to B cells producing anti-TA antibodies. Thus, a key step in increasing the immunogenicity of TAs of tumor cells within the individual patient is the induction of robust uptake of many tumor cells and cell membranes by APCs. 

The robust in situ uptake of TAs by APCs can be achieved by harnessing of the natural anti-Gal antibody. Since anti-Gal is ubiquitously produced in large amounts in all humans who are not severely immunosuppressed, it was hypothesized that cancer patients may benefit from vaccination with lethally irradiated self-tumor cells or cell membranes engineered to present α-gal epitopes [52,53]. In addition, it was suggested that self-TAs in tumor lesions could be converted in situ into personalized anti-TA vaccines by inducing expression α-gal epitopes on the tumor cells within treated lesions [54]. The mechanism for such an in situ conversion of TAs into a vaccine is illustrated in Figure 1A, according to the following steps: Step 1—Vaccinating tumor cells presenting α-gal epitopes bind the natural anti-Gal IgM and IgG antibody molecules and activates the complement system. Step 2—In addition to tumor cell cytolysis by anti-Gal-mediated CDC, the chemotactic complement cleavage peptides C5a and C3a, generated by complement activation, effectively recruit macrophages and dendritic cells to the vaccination site. Step 3—The Fcγ receptors (FcγR) on these recruited APCs bind Fc “tails” of the opsonizing anti-Gal. This Fc/FcγR interaction induces rearrangements in the actin cytoskeleton of the APC that lead to the phagocytosis of the opsonized tumor cells and cell membranes [55]. The binding of C3b on the opsonized cells to C3b receptors (CR1) on APCs induces similar actin cytoskeleton rearrangements and the phagocytosis of the cells presenting C3b by the APCs [56]. Step 4—Due to this robust phagocytosis, many internalized tumor cells and cell membranes are transported by the APCs to regional lymph nodes. TAs on internalized tumor cell membranes are processed by the APCs and TA peptides with high affinity to HLA molecules are presented by these molecules for the effective activation of TA-specific T cells. The activated T cells mount protective humoral and cellular immune responses against metastatic tumor cells presenting the immunizing TA.

The principle of an increased immunogenicity of vaccines, by targeting them as immune complexes for robust phagocytosis by APCs, is not unique to tumor vaccines immunocomplexed with anti-Gal; it has been observed with a variety of antigens used as vaccines. An increase of 10–1000 folds in antibody titers following immunization with antigens immunocomplexed with their corresponding antibodies (other than anti-Gal) in comparison with immunization by the antigen alone was found with several vaccinating antigens, including Tetanus toxoid [57,58,59], hepatitis B antigen [60,61], and the simian immunodeficiency virus (SIV) Gag antigen [62]. The extent of increased immunogenicity of vaccines presenting α-gal epitopes and immunocomplexed in situ by anti-Gal could be quantified with influenza virus vaccine [63] and gp120 of the HIV vaccine, presenting α-gal epitopes [64] in anti-Gal-producing mice. The presentation of α-gal epitopes in both vaccines resulted in ~100-fold higher titers of protective anti-viral antibodies and in much better protection against these viruses than following vaccination with the corresponding vaccines lacking this epitope.

The high uptake efficacy of self-tumor cells presenting α-gal epitopes by APCs could be demonstrated in vitro by the co-incubation of B cell lymphoma cells from a patient with autologous macrophages or dendritic cells and in the presence of an autologous anti-Gal antibody [65]. The lymphoma cells were enzymatically engineered to present α-gal epitopes [52,65]. Cells were incubated for 2 h at 37 °C with macrophages and dendritic cells in the presence of anti-Gal. Within this period, as many as eight tumor cells opsonized by anti-Gal were internalized by macrophages (Figure 1B). This phagocytosis was found to be mediated by the FcγR1 (CD64) on the human APC [65]. In contrast, tumor cells lacking α-gal epitopes were not phagocytosed by macrophages. Dendritic cells also internalized tumor cells opsonized by anti-Gal, albeit less than macrophages, whereas no phagocytosis was observed with lymphoma cells lacking α-gal epitopes (Figure 1B). A similar robust anti-Gal-mediated uptake by macrophages of immunizing cells presenting α-gal epitopes could be demonstrated with human leukemia cells [52].

**Figure 1 pharmaceutics-16-01263-f001:**
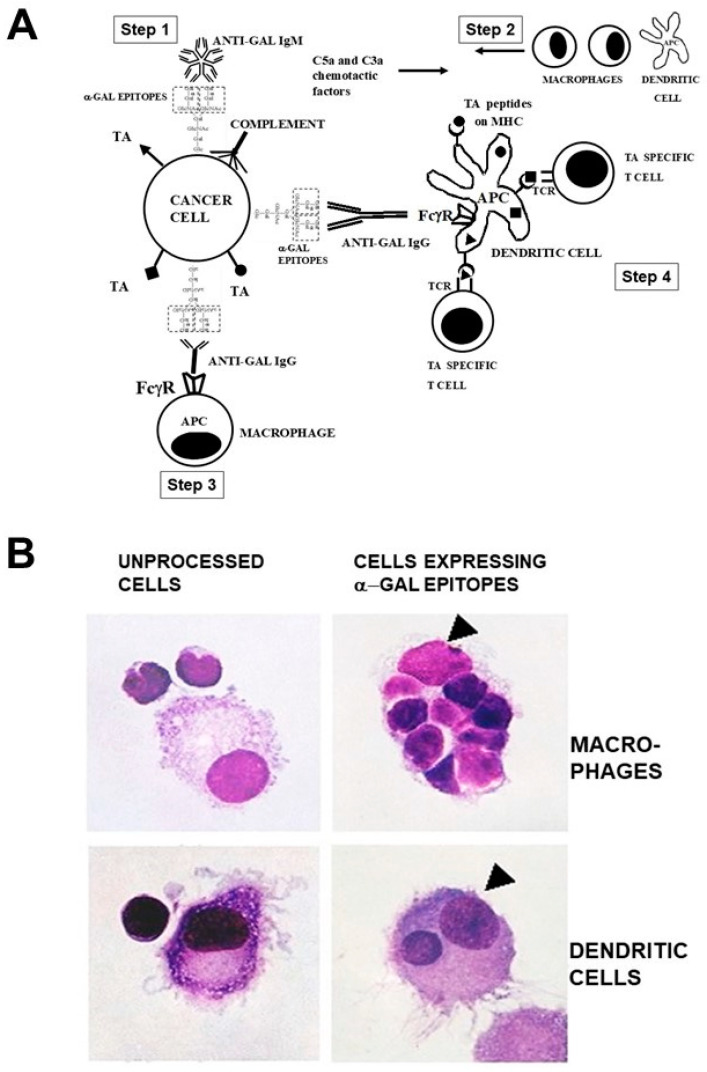
Anti-Gal binding to α-gal epitopes on tumor cells targets them and their TAs for uptake by APCs and improves TA immunogenicity. (**A**) An illustration of the hypothesis. Step 1—Tumor cells presenting α-gal epitopes (dashed-line rectangles) bind natural anti-Gal IgM and IgG antibodies. Step 2—The activated complement system generates complement cleavage chemotactic peptidesC5a and C3a, that induce the rapid recruitment of APCs. Step 3—Anti-Gal IgG opsonized tumor cells are phagocytosed by APCs following Fc/FcγR interaction. Step 4—The internalized TAs (●, ■, ▲) are transported by the APCs to regional lymph nodes and are processed. Immunogenic TA peptides are present on class I and class II MHC molecules, thereby activating tumor specific cytotoxic and helper T cell clones with the corresponding T cell receptors (TCRs). The detailed structure of the glycolipid with the α-gal epitope is presented in Figure 3A at a larger magnification. (**B**) The anti-Gal-mediated uptake by autologous APCs of human B lymphoma cells engineered to present α-gal epitopes. Lymphoma cells with or without α-gal epitopes were incubated with autologous anti-Gal and autologous macrophages or dendritic cells. Triangles mark the nuclei of the APCs. Note the robust phagocytosis of lymphoma cells presenting α-gal epitopes by the macrophage and of one lymphoma cell phagocytosed by the dendritic cell. No uptake of lymphoma cells lacking α-gal epitopes is observed (Hematoxylin and Eosin [H&E] ×1000). Modified with permission from [65] 2005, Haematologica.

## 4. Protective Efficacy of Self-TA Vaccines Presenting α-gal Epitopes

### 4.1. The Experimental Animal Model

Studies on the efficacy of α-gal-presenting TA vaccines cannot be performed in standard experimental animal models, such as mouse, rat, rabbit or guinea-pig. The reason for this is that these animals, like other nonprimate mammals, naturally synthesize the α-gal epitope, and thus are immunotolerant to it and cannot produce the anti-Gal antibody. This obstacle was overcome by the generation of knockout mice for the α1,3GT gene *GGTA1* (GT-KO mice) [66,67]. These mice lack α-gal epitopes and can produce the anti-Gal antibody following immunization with xenogeneic cells or cell membranes presenting α-gal epitopes, such as porcine mononuclear cells [68], porcine kidney membrane homogenate [46,47], rabbit red cells [69] or immunization with glycoproteins such as synthetic α-gal linked to bovine serum albumin (α-gal-BSA) [70]. The choice for tumor cells in such studies is also limited, because most mouse cell lines naturally present α-gal epitopes and therefore are lysed by anti-Gal in the presence of a complement or by ADCC. One of the very few mouse cancer cell lines lacking the α-gal epitope is B16 melanoma, which grows well in GT-KO mice and does not bind anti-Gal [69]. 

### 4.2. Immune Protection against B16 Melanoma by Tumor Cell Vaccines Presenting α-gal Epitopes

A subclone of B16 melanoma called “B16/BL6” (referred to here as B16), which is highly tumorigenic, was used in the preliminary studies on the efficacy of tumor vaccines presenting α-gal epitopes [69]. These cells were engineered to synthesize and present α-gal epitopes by stable transfection with the α1,3galactosyltransferase gene *GGTA1*. This transgenic cell line, designated B16_α-gal_, expressed multiple α-gal epitopes and bound the anti-Gal antibody [69]. B16_α-gal_ cells were lethally irradiated and 2 × 10^6^ of these cells were used as a vaccine, injected subcutaneously into anti-Gal-producing GT-KO mice (i.e., mice receiving prior immunizations with rabbit red cells). Injections of 2 × 10^6^ irradiated B16 cells (lacking α-gal epitopes) were used as the vaccine control. Two weeks post immunization, the mice were challenged subcutaneously with 0.5 × 10^6^ live B16 cells, and tumor development was monitored for 60 days. A third of the mice immunized with B16_α-gal_ cells did not develop any tumor after the challenge, whereas all mice immunized with the parental B16 cells developed tumors within 8–26 days post challenge (Figure 2A). These findings strongly suggested that one immunization with irradiated engineered B16_α-gal_ tumor cells elicited a protective immune response against the TAs of the tumor, as indicated by the destruction of the challenging B16 cells. In contrast, vaccines consisting of the same cells which lacked α-gal epitopes failed in eliciting a protective anti-TA immune response [69].

Of particular interest were the histological characteristics of the developing tumors. Tumors developing in mice immunized with B16 cells displayed highly proliferative tumor cells, as indicated by the relatively small size of the cells and basophilic staining of the cytoplasm due to the high concentration of ribosomes (Figure 2B). In contrast, tumors developing in mice vaccinated with B16_α-gal_ cells were surrounded by multiple mononuclear cells (Figure 2C). An immunostaining analysis indicated that ~70% of these mononuclear cells were T cells stained by anti-CD3 antibodies, and 30% were macrophages [69]. No B cells were found in these infiltrates. Tumor cells adjacent to the mononuclear cell were large and melanin-producing, with vacuolated cytoplasm, suggesting ongoing cytolysis. The melanin (dark granules) in these cells suggested that they stopped proliferating and matured into melanin-producing cells, due to the cytokines produced by infiltrating T cells and macrophages. This α-gal therapy in mice immunized with B16_α-gal_ was subsequently independently validated using B16_α-gal_ cells that were generated by stable transfection with a retrovirus vector containing the α1,3GT gene *GGTA1* [71]. Similar protective results were also obtained by generating B16_α-gal_ cell vaccines with adenovirus containing this gene (called AdαGT), which was transduced into B16 cells [72].

The induction of a protective anti-TA immune response was also observed in GT-KO mice immunized with both pancreatic carcinoma cells [73] and ovarian carcinoma cells engineered to present α-gal epitopes [74], as well as with breast cancer cells engineered to present α-gal epitopes and fused with dendritic cells [75]. All these studies have suggested that vaccines made of tumor cells engineered to present α-gal epitopes can induce a protective immune response against self-TAs without the need for identifying these TAs in the individual patient. This immune response can destroy metastatic cells and slow or eliminate the race between the expanding tumor mass and the immune-response-destroying tumor cells. However, as may be inferred from Figure 2C, if the vaccine is applied in advanced stages of the disease when the tumor mass is large, the protective immune response against the various TAs may not be effective enough to destroy all the tumor cells but may be able to slow the progression of the disease. 

### 4.3. Clinical Trials with Tumor Cells Engineered In Vitro to Present α-gal Epitopes

The α-gal immunotherapy with autologous tumor cells and cell membranes engineered to present α-gal epitopes was studied in Phase I clinical trials with refractory patients at an advanced stage of the disease, having hepatocellular carcinoma [76], pancreatic carcinoma [77] and lymphoma [78]. Homogenates of cell membranes from the two solid tumors [76,77] and lymphoma cell suspensions [78] were subjected to the enzymatic synthesis of α-gal epitopes by recombinant α1,3GT, UDP-Gal and neuraminidase, as previously detailed [52,53]. The tumor membrane homogenates and lymphoma cells presenting α-gal epitopes were incubated for several days with autologous dendritic cells and anti-Gal, in order to target the vaccinating materials for robust uptake by APCs. Subsequently, the whole mixture was administered by several injections of autologous vaccine to the patients. Among the hepatocellular carcinoma patients, an average of a 70% increase in survival time and increase in positive delayed hypersensitivity were observed, with no serious side effects or autoimmune diseases [76]. Treated patients with pancreatic carcinoma [77] also displayed a strong delayed-type hypersensitivity to the autologous cancer cell lysate with no serious side effects. Among the fourteen patients with lymphoma, complete and partial remission occurred in four and three patients, respectively [78]. The disease status remained unchanged in five patients, and disease progression was observed in two patients. Overall, these studies suggested that the vaccination with of self-tumor cells or cell membranes presenting α-gal epitopes is safe, and that this vaccination process may induce a protective immune response in some of the treated patients. However, at advanced stages of the disease, the efficacy of this immunotherapy may be limited because of the large tumor mass in the treated patient.

## 5. Conversion of Self-TAs into Vaccines by Natural α-gal Micelles

The in vitro engineering of leukemia and lymphoma cells to express α-gal epitopes and to serve as a self-TA vaccine is relatively easy and involves harvesting tumor cells from blood or lymphoid organs and the use of recombinant α1,3GT, UDP-Gal, and neuraminidase [52,53] or AdαGT transduction [72,79]. However, the technical difficulties involved in preparing α-gal vaccines against self-TAs by homogenizing resected solid tumors and synthesizing α-gal epitopes on the fragmented cell membranes prompted the development of methods for the in situ induction of α-gal epitope expression on the cells of solid tumors. Tumor cells induced within lesions to present α-gal epitopes may convert into anti-self-TA vaccines, functioning as described in Figure 1A. Studies on the expression of α-gal epitopes within the treated lesions following an intra-tumoral injection of recombinant α1,3GT, AdαGT, or α-gal micelles indicated that α-gal micelles had the highest efficacy for achieving the intra-tumoral expression of α-gal epitopes [54].

### α-gal Glycolipids and α-gal Micelles

α-gal micelles are micelles comprised of α-gal glycolipids (Figure 3A). These glycolipids are made of glycans with terminal α-gal epitopes, which are linked at the reducing end to a ceramide composed of sphingosine and fatty acid joined by an amide bond. When α-gal glycolipids are suspended in an aqueous solution, they form small spheric structures called micelles, in which hydrophobic chains of the ceramides form the inner area of the sphere, whereas the hydrophilic glycans protrude into the aqueous solution surrounding the micelle (Figure 3B). Individual α-gal glycolipid molecules detach in a reversible manner from the micelle and are inserted back into it. However, when α-gal micelles are near cell membranes, the α-gal glycolipid molecules detaching from the micelles insert in an irreversible manner into the lipid bilayer of the cell membranes. The reason for this irreversible insertion is that, energetically, the ceramide “tails” of glycolipids are much more stable when surrounded by phospholipids of the cell membrane than in the micelles, surrounded by water molecules. Therefore, the injection of α-gal micelles into solid tumors results in their rapid insertion into the tumor cell membranes (Figure 3B), leading to the presentation of multiple α-gal epitopes on the tumor cells. These epitopes bind anti-Gal and initiate the sequence of processes illustrated in Figure 1A. 

### 5.2. Production of α-gal Micelles

A rich source for natural α-gal glycolipids is the membranes of rabbit red cells, which express ~2 × 10^6^ α-gal epitopes per cell, many of which are of glycolipids and the rest are of glycoproteins [29,30,33]. The glycolipid example in Figure 3 has 10 carbohydrate units with two branches (antennae), each with an α-gal epitope. The α-gal glycolipids in rabbit-red-cell membranes were previously shown to have a total of 1–8 α-gal epitopes per molecule, each on one branch [80,81,82,83,84,85]. As detailed in ref. [54], the identification of extracted α-gal glycolipids from rabbit-red-cell membranes was performed by thin-layer chromatography analysis (TLC). This analysis indicated that the glycolipids were extracted together with phospholipids and cholesterol by the incubation of rabbit-red-cell membranes (ghosts) in a solution of chloroform–methanol 1:2. The glycolipids were further purified by Folch partition which removes the phospholipids and cholesterol [54,86]. The staining of the purified glycolipids chromatographed on TLC plates with a human anti-Gal antibody indicated that they had the size of 5, 7, 10, 15, 20 and 25 carbohydrate units (hexosides) with corresponding 1, 1, 2, 3, 4, and 5 antennae, and that each of these antennae was capped with the α-gal epitope [54]. Because of the large size of glycolipids with >25 carbohydrate units, their bands could not be separated from that of the glycolipid with 25 carbohydrate units (ceramide eikosipente hexoside) [54].

**Figure 3 pharmaceutics-16-01263-f003:**
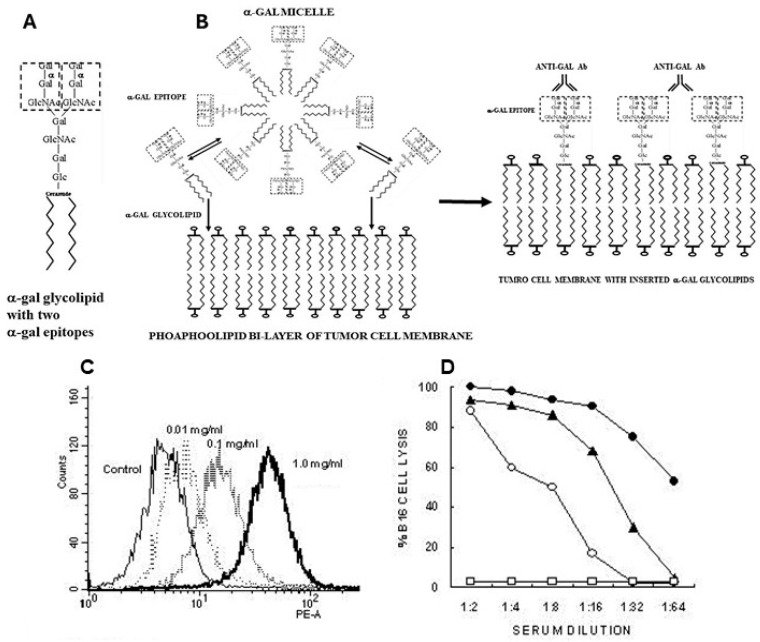
α-Gal glycolipids in micelles are inserted into tumor cell membranes. (**A**) A representative α-gal glycolipid-ceramide decahexoside (10 carbohydrate units) with two branches, each carrying an α-gal epitope (marked by a dashed rectangle). (**B**) Illustrated section in an α-gal micelle sphere comprised of α-gal glycolipids. These glycolipids detach from the micelle and re-insert into it. However, if the micelles are near tumor cell membranes, the α-gal glycolipids spontaneously insert into these membranes in a stable manner, resulting in the subsequent binding of anti-Gal to α-gal epitopes on these glycolipids. (**C**) The binding of monoclonal anti-Gal antibody to B16 cells following the incubation of these cells for 2 h at 37 °C with various concentrations of α-gal micelles, as measured by flow cytometry. (**D**) Complement dependent cytolysis (CDC) of B16 cells incubated for 2 h with 1.0 mg/mL α-gal micelles. Mouse serum containing anti-Gal and rabbit complement (●); mouse serum without rabbit complement (○); fresh human serum (▲); control B16 cells lacking α-gal epitopes and incubated with mouse serum and rabbit complement, or with human serum (☐) (mean of four experiments). Modified with permission from ref. [87] 2018, Elsevier.

### 5.3. Insertion of α-gal Glycolipids into Tumor Cell Membranes

The insertion of α-gal glycolipids into tumor cells incubated with α-gal micelles is demonstrated in Figure 3C. B16 melanoma cells in suspension were incubated for 2 h at 37 °C with α-gal micelles at various concentrations of the micelles, then washed and immunostained with the monoclonal anti-Gal antibody M86, which binds specifically to α-gal epitopes [88]. Flow cytometry analysis demonstrated a correlation between the concentration of the α-gal micelles and the extent of α-gal glycolipids inserted into the B16 cell membranes [54]. The binding of serum anti-Gal to inserted α-gal glycolipids could further mediate complement dependent cytolysis (CDC) of the B16 cells (Figure 3D).

### 5.4. In Vivo Recruitment of APC into Treated Tumor Lesions

The in vivo effects of intra-tumoral injection of α-gal micelles were determined with B16 melanoma lesions generated in the skin of anti-Gal-producing GT-KO mice. Tumor lesions were generated in mice that were injected intradermally in two sites ~2.5 cm apart, with 1 × 10^6^ live B16 cells. Within 5 days, the cutaneous tumor lesions developed with a diameter of ~5.0 mm. One of the lesions in each mouse was injected at three points with 1.0 mg α-gal micelles, and the other was injected with PBS as control. No infiltration of PMN or mononuclear cells was observed in the histologic sections of PBS-injected tumors 2 and 7 days post injection (Figure 4A). In contrast, the extensive infiltration of mononuclear cells was observed within 48 h post injection of α-gal micelles (Figure 4B). This infiltration, mediated by the complement cleavage chemotactic peptides, further increased after 7 days (Figure 4C) [54]. An analysis of the tumor-infiltrating mononuclear cells in the α-gal micelle-injected lesions indicated that they include dendritic cells, macrophages, NK cells, CD4+ and CD8+ T cells [54]. An inspection of the tumors 10 days after these injections indicated that the PBS injected tumors increased their size ~3 fold (Figure 4D,E). In contrast, most tumors injected with α-gal micelles either stopped growing or decreased in size, some to the extent of near-complete elimination (Figure 4D,E). These findings implied that the in vivo insertion of α-gal glycolipids into the B16 cell membranes resulted in the complement activation and killing of varying proportions of the tumor cells within the treated lesions. Repeating this study in wild-type mice (WT mice) demonstrated no significant differences between α-gal micelle-injected and PBS-injected lesions (Figure 4E) [54]. This further implied that in the absence of anti-Gal (WT mice synthesize α-gal epitopes and cannot produce this antibody), the α-gal micelle injection has no effect on the tumor growth. It should be stressed that killing of tumor cells within lesions injected with α-gal micelles is not the main objective of this therapy. The main objective is to convert live or dead tumor cells into cells that are “palatable” to APCs by their opsonization with anti-Gal. The robust phagocytosis of anti-Gal opsonized tumor cells by the recruited APC achieves the aim of internalization of large amounts of TAs into the APCs without the need for identifying the TAs in the individual patient.

### 5.5. α-gal Micelles Mediated Increased Transport by APCs of Processed Surrogate TA Peptides

The increase in uptake and processing of TAs, mediated by intra-tumoral injection of α-gal micelles, could be demonstrated with B16 cells containing the ovalbumin (OVA) transgene and producing OVA as a surrogate TA (designated B16/OVA cells) [90]. The detection of APC-internalizing B16/OVA cells was performed by their coincubation with the CD8+ hybridoma B3Z T cells, which have the T cell receptor (TCR) for the presented peptide SIINFEKL [91,92,93]. This peptide is the immunodominant OVA peptide when presented on class I MHC molecules.

The extent of the uptake of B16/OVA cells by APCs within treated tumor lesions can be evaluated by the presentation of the peptide SIINFEKL on MHC molecules on the APCs, since the internalized OVA is processed within the APCs by being converted to peptides, and the immunodominant SIINFEKL peptide is transported on MHC molecules and presented on the cell membrane of the APC. When TCRs on B3Z hybridoma T cells engage the peptide SIINFEKL presented on class I MHC molecules of APCs, the B3Z cells are activated. There is a correlation between the extent of uptake of B16/OVA cells by APCs and the subsequent activation of B3Z cells when the two cell types are co-incubated in vitro [91,92,93]. Within the activated B3Z cells, their β-galactosidase transgene (*Lac-Z*) with IL2 promoter is activated. This activation simulates the activation of CD8+ T cells, which results in the production of IL2 following activation by the corresponding peptide. The catalytic activity of the produced β-galactosidase is detected by the fluorescence of a substrate cleaved by this enzyme. This fluorescence can be detected in the activated B3Z cells by flow cytometry.

For analysis of the phagocytosis of OVA by APCs, B16/OVA lesions with a diameter of ~5 mm on the right thigh of anti-Gal-producing GT-KO mice were injected on day 1 and day 7 with 1.0 mg α-gal micelles or with PBS. The inguinal lymph nodes (i.e., draining lymph nodes of the lesion) in the right thigh and those in the left thigh were harvested on day 14 and their cells were co-incubated with B3Z cells. The proportion of APC-presenting SIINFEKL as a surrogate TA peptide, which migrated from the treated lesion to the draining lymph nodes, could be deduced from the percentage of activated B3Z cells. The proportion of SIINFEKL-presenting APC in the right leg inguinal lymph nodes draining the B16/OVA lesion injected with α-gal micelles, was several-fold higher than that in the non-draining inguinal lymph nodes in the left leg of the mice. In addition, the proportion of APC-presenting SIINFEKL in lymph nodes draining α-gal micelle-treated lesions was much higher than that of similar APCs in lymph nodes draining PBS-treated tumors (Figure 5) [54]. These findings strongly suggested a robust uptake of OVA as surrogate TAs in B16/OVA cells, as well as a higher transport and processing of the OVA peptide SIINFEKL by APC in α-gal micelle-injected lesions in comparison to PBS-injected control lesions.

### 5.6. Increased Protection against Distant Metastatic Cells by Intra-Tumoral Injection of α-gal Micelles

The efficacy of α-gal micelles in inducing a systemic protective immune response which prevents the development of distant “established metastases” (abscopal effect) was evaluated in anti-Gal-producing GT-KO mice by two methods. First, the mice were injected subcutaneously with live 1 × 10^6^ B16 melanoma cells in the right flank (simulating the “primary tumor”). When the tumors reached ~5 mm (after 5 days) the tumors were injected with 1.0 mg α-gal micelles in PBS. One and two weeks later, the mice received two additional injections of 1.0 mg α-gal micelles. Control mice received three injections of PBS instead of α-gal micelles. One week after the third injection, the mice were challenged with 0.5 × 10^6^ live B16 melanoma cells in the left flank, thereby simulating distant metastatic cells. Ten of the fifteen mice treated with α-gal micelles displayed no growth of the challenging tumor cells into a melanoma lesion (Figure 6A). In contrast, the challenging tumor cells in all PBS-treated mice developed into melanoma lesions (Figure 6B) [94]. 

In a second experiment, simultaneously with the injection of 1 × 10^6^ B16 melanoma cells in the right flank as “primary tumor”, the mice were injected with 1 × 10^4^ tumor cells in the left flank, which simulated a “distant micro-metastasis”. The primary tumor lesions reaching the size of ~5 mm were injected with 1.0 mg α-gal micelles, or with PBS as control. The injections were repeated one week later and the development of the “distant metastasis” in each of the mice was monitored. In this experiment, distant micro-metastases were established for 5 days before the primary tumor was injected with α-gal micelles. In all PBS-treated mice, “distant metastasis” developed into visible 2–8 mm tumor lesions in the left flank by day 15 after the first injection of the “primary” tumor (Figure 6D). However, in 50% of mice injected with α-gal micelles into the “primary tumor”, no tumor growth was observed at the “distant metastasis” site, even after 30 days. In the rest of the mice in this group, slower tumor growth was observed in comparison to most PBS-treated mice (Figure 6C) [94]. These studies indicated that the injection of the melanoma lesions with α-gal micelles resulted in the development of a systemic anti-TA immune response, which protected against distant metastases growth in a significant proportion of the treated mice.

### 5.7. CD8+ T Cells Are the Main Protective Cells in Immunized Mice

The identification of the immune cells which mediate the immune protection against the development of distant B16 melanoma micrometastases into tumor lesions was evaluated by adoptive transfer studies of splenocytes from tumor-bearing mice treated with α-gal micelles. B16 tumor 5 mm lesions in GT-KO mice producing anti-Gal were injected twice with 1.0 mg α-gal micelles in one-week intervals. The spleens of the tumor-bearing mice were harvested one week after the second injection. Forty million of the harvested splenocytes were administered intravenously to naïve GT-KO mice that were also challenged 24 h earlier with 0.5 × 10^6^ live B16 cells. The rest of the splenocytes were subjected to the depletion of CD8+ T cells using magnetic micro-beads coated with anti-CD8+ antibody. Forty million of the splenocytes depleted of CD8+ T cells were administered to the paired naïve mice challenged with B16 cells 24 h earlier [94]. As shown in Figure 7, the transferred splenocytes prevented the development of challenging B16 cells into melanoma lesions in most of the recipients. However, all paired recipients of splenocytes depleted of CD8+ T cells displayed growth of tumors that reached the size of 25 mm within ≤30 days. These findings clearly indicated that the main cell population which mediated the destruction of challenging B16 cells in the naïve recipients was the TA-specific CD8+ T cells that were activated following the injection of α-gal micelles into the tumor lesions of the donor mice [94].

Overall, studies with α-gal micelles in the pre-clinical model of anti-Gal producing GT-KO mice strongly suggested that injection of these micelles into solid tumors can convert them into in situ vaccines against TAs on the treated tumor cells, without the need for identifying the TAs. Such vaccines may induce a protective immune response against metastatic tumor cells that present TAs in the treated patient. In the absence of α-gal micelle treatment, the patient’s TAs may be “ignored” by the immune system because of the insufficient uptake and processing of TAs by APCs and thus, resulting in poor presentation of the immunogenic TA peptides for T cell activation.

## 6. Clinical Trials with Natural α-gal Micelles

Based on the studies described above, two Phase 1 clinical trials were performed with α-gal micelles. The first was a dose escalation and safety study in patients with solid tumors at advanced stages of the disease [95]. The second trial had the same objectives and was performed in patients with advance cutaneous melanoma [96]. All patients were treated with standard therapy and experienced a recurrence of the disease prior to each of the clinical trials.

In the first study, 11 patients with a variety of solid tumors were treated at the UMass Medical Center after approval by a local Human Investigations Committee (IRB) and in accordance with an assurance approved by the Department of Health and Human Services, (FDA IND 12946). Eight patients had metastatic cancers, including colon, neuroendocrine, prostate, renal, ovarian, and mucinous appendiceal tumors [95]. The three patients who did not have metastatic disease had locally advanced pancreatic adenocarcinoma. The patients received intra-tumoral 0.1, 1.0, or 10 mg α-gal micelles injected into the target tumor lesions under ultrasound guidance, CT guidance, or under manual and visual control in patients who had large palpable tumors. The patients were monitored at regular intervals for a period of two years. None of the patients developed clinical or laboratory signs of toxicity, symptoms of allergic responses, autoimmune conditions, anti-nuclear antibodies, or antibodies to normal tissue antigens [95]. The injected tumors did not regress, and patients developed evidence of disease progression ranging from minimal to significant at various time points after the 4-week endpoint for toxicity analysis. However, several patients were alive with disease for 13+ to 48+ months, even though disease progression was evident by imaging. In addition, two patients with advanced pancreatic adenocarcinoma, who prior to the treatment were expected to survive only 4–6 months, survived following the α-gal micelle treatment for 18 and 23 months [95].

A second study was performed at the University of Wisconsin Carbone Cancer Center under FDA IND 12946, in nine patients with unresectable melanoma (recurrent stage III or stage IV) that was refractory to therapy, with at least one cutaneous metastasis [96]. All patients received intra-tumoral two injections of 0.1, 1.0, or 10 mg α-gal micelles, administered 4 weeks apart. Prior to the second injection, the patients received 10 μg of α-gal micelles intradermally and were then observed for one hour to confirm the absence of α-gal syndrome. The sizes of treated and untreated lesions were monitored during regular clinic visits. Two patients maintained stable disease for 7 and 8 months after treatment, and the remaining seven patients had disease progression. Three of the injected lesions were stable in size, 4 weeks after the second α-gal micelle injection, and the remaining six lesions had at least a 20% increase in that period. However, several findings suggested some response to the treatment. An increased necrosis of the tumor tissue was demonstrated in biopsies of the injected lesions in five of nine patients, and in nontreated tumor lesions in two of four evaluable patients. In addition, in three of eight patients, an approximate two-fold to five-fold increase in the frequency of TA pentamer+ cells were observed post treatment vs. pre-treatment [96].

Overall, these two Phase I clinical trials indicated that the treatment was safe; however, the treatment in the advanced stages of the disease did not induce a potent enough protective immune response to result in the complete cure or in the distinct regression of tumors. It is possible that increased doses may be more effective in eliciting a protective immune response in patients with a large tumor burden. In addition, it is reasonable to assume that the treatment with α-gal micelles is likely to be more effective at early stages of the disease, when the tumor burden is lower, and thus, the probability of the immune system for “catching up” with the developing metastases is higher. It is also important to administer this treatment to patients in whom the immune system is functional and not impaired by recent chemotherapy treatments. 

## 7. Conversion of Self-TAs into Vaccines by Synthetic α-gal Micelles

The induction of a protective immune response against distant metastases in mice with B16 melanoma lesions injected with α-gal micelles [54,94], prompted the biotech companies Agalimmune Ltd. and BioLineRx Ltd. to determine whether a similar conversion of tumors into a vaccine against self-TAs is feasible with synthetic α-gal glycolipids [42]. For this purpose, a fully synthetic α-gal glycolipid-like molecule, called “AGI-134” was chosen (produced by Kode Biotech, Auckland, NZ). AGI-134 comprises a synthetic α-gal epitope linked via an adipate (di-carboxylate) linker to a lipid tail [97].

As detailed in ref. [42], in vitro studies with AGI-134 indicated that it can insert well into the cell membrane of both human and mouse tumor cells and bind human and mouse anti-Gal antibodies. This binding further activated the complement cascade and induced CDC as well as ADCC. AGI-134-treated B16 cells were opsonized by anti-Gal and phagocytosed by APCs. In addition, ovalbumin (OVA), serving as a surrogate TA in cells treated with AGI-134, was effectively processed following uptake by APCs, and its immunodominant peptide SIINFEKL was cross-presented by the APCs [42]. 

Studies on the efficacy of AGI-134 were further performed in vivo in anti-Gal-producing GT-KO mice. The intra-tumoral injection of AGI-134 into B16 melanoma lesions in these mice resulted in regression of the subcutaneous tumors, similar to the regression observed with natural α-gal micelles in Figure 4 above. Moreover, in most mice, the injection of AGI-134 into “primary” tumors induced a protective immune response that prevented the development of distant metastases, simulated by challenging 1 × 10^4^ B16 cells injected into the contralateral flank [42]. No protective immune response was observed if AGI-134 was administered into B16 lesions in mice that lacked the anti-Gal antibody. The efficacy of AGI-134 was found to be dose-dependent. The maximal efficacy of distant metastases’ prevention was achieved with 1.0 mg AGI-134. At suboptimal doses of 0.5 mg and 0.1 mg, the efficacy was reduced; however, significant protection was observed in comparison to PBS controls. It is of interest to note that AGI-134 at suboptimal doses synergized with the check-point inhibitor anti-PD-1 antibody in preventing development of distant metastases [42].

Overall, the studies with AGI-134 synthetic micelles suggested that these micelles display characteristics similar to those of natural α-gal micelles, in that their injection into tumors results in the conversion of the treated lesions into vaccines that induce a protective immune response against self-TAs, thereby preventing metastases’ formation.

## 8. Clinical Trial with Synthetic α-gal Micelles

The pre-clinical studies with AGI-134 described above [42] were followed by a multicenter open-label Phase 1/2a clinical trial performed by BiolineRX Ltd., in order to evaluate the safety and efficacy of AGI-134 immunotherapy. The study was performed with 38 patients who had a variety of solid tumors in the UK, Spain and Israel. This clinical trial was listed as ClinicalTrials.gov ID NCT03593226. The results of this trial were announced so far only as a press release by BioLineRX on 20 December 2022 [98]. That press release indicated that after an accelerated dose-escalation study in 5 patients to determine the maximum tolerated dose, the safety and tolerability of AGI-134 were evaluated in 33 patients. The treatment with AGI-134 was found to be well tolerated, and the adverse events were mostly transient, and mild to moderate in their severity. Clinical response evaluation according to RECIST1.1 criteria indicated that a stable disease (SD) was observed in 11 of the 38 patients (29%), and that 7 of these 11 patients had a prior checkpoint-inhibitor therapy, which failed. Biopsy studies indicated that, following treatment with AGI-134, leukocyte infiltration, including macrophages and dendritic cells, was observed within a few days in 59% of the treated tumors. One of the most significant indications for an anti-tumor immune response following AGI-134 treatment has been the infiltration of CD4+ T helper cells and of CD8+ cytotoxic T cells in 47% of the un-injected tumor lesions. These findings support the assumption that α-gal glycolipids inserted into tumor cell membranes bind anti-Gal and target these cells and cell membranes for robust uptake by APCs. These APCs transport processed TA peptides to lymph nodes for the activation of TA-specific CD4+ and CD8+ T cells.

## 9. Advantages, and Limitations

### 9.1. Neoadjuvant Therapy as a Personalized Vaccine—Advantages

The majority of presently used cancer immunotherapies, such as check-point inhibitor antibodies, and vaccines of neoantigens and of shared tumor antigens, are usually used for the treatment of patients at advanced stages of the disease, with the aim of activating the immune system to react against the TAs of the individual patient [99]. The α-gal micelle immunotherapy provides a unique opportunity to induce a protective immune response against the full range of TAs on the metastatic cells of the individual patient right upon the detection of the primary tumor and without the need for identifying the TAs. In many patients, the resection of tumors is performed several weeks after detection. Therefore, it is suggested that intra-tumoral injection of α-gal micelles may be performed upon detection of the tumor. Studies in the mouse model indicated that the recruitment of APCs into the treated lesion, the uptake of anti-Gal opsonized tumor cells and cell membranes by the recruited APCs, and migration of such APCs out of the tumor to lymph nodes may occur within 1–3 weeks post treatment. Thus, at the time of the treated lesion resection, the APCs are likely to be already in the course of transporting processed TA peptides to lymph nodes for eliciting a protective immune response against metastatic cells presenting these TAs. Tumor cell proliferation in humans is usually much slower than that of the B16 melanoma cells in mice, which double their size every 4–8 days. This provides more time for primed TA-specific T cells to “find” and destroy metastatic cells. Therefore, future α-gal micelle immunotherapy may be beneficial in patients with micro-metastases which are not visible at the time of the primary tumor detection. Furthermore, the efficacy of the immune system at the time of detection is likely to be optimal for induction of an anti-TA response, since it was not yet affected by any chemotherapy treatment.

### 9.2. Amplifying the Efficacy of Checkpoint-Inhibitor Treatment—Advantages

The pre-clinical and clinical studies described above indicated that α-gal micelle treatments in cancer patients can result in the amplification of TA-specific T cell activation, which contributes to protection against metastases’ development. In addition, this treatment can synergize with the various checkpoint-inhibitor treatments currently used in cancer immunotherapy. The success of the checkpoint-blockade treatment depends on the amounts of previously primed TA-specific T cells that infiltrate tumor lesions and are blocked from killing the tumor cells presenting the corresponding TAs [99]. The treatment of patients with α-gal micelles prior to checkpoint-blockade treatment may increase the number of primed TA-specific T cells; therefore, many more tumor-infiltrating T cells will be able to function as cytotoxic T cells than those in patients that were not treated by α-gal micelles. This treatment is likely to be of particular benefit for patients in whom the TA are relatively “weak”, as implied from the significant proportion of cancer patients who presently fail the checkpoint-blockade treatment [11]. This assumption is supported by the observations on the synergism between the anti-PD1 antibody and synthetic α-gal micelles in preventing metastases’ development in anti-Gal-producing mice carrying B16 melanoma [42] and may be further supported by future clinical trials combining the two treatments. 

### 9.3. Immunotherapy of Hematological Malignancies with α-gal Micelles—Advantages 

Immunotherapy with α-gal micelles may also be applicable in patients with hematological malignancies. The spontaneous insertion of α-gal glycolipids following the in vitro incubation of B16 melanoma cells with α-gal micelles was demonstrated above in Figure 3C,D. A similar in vitro incubation of lethally irradiated human leukemia, lymphoma or myeloma cells with α-gal micelles at 37 °C will result in the insertion of α-gal glycolipids into tumor cell membranes and the presentation of multiple α-gal epitopes on the tumor cells. The immunization of the treated patient by such personalized anti-TA vaccines presenting α-gal epitopes may lead to the activation of the immune system against self-TAs according to the steps described in Figure 1A. A similar amplified immune response was observed following immunization with α-gal-presenting inactivated influenza virus [63] and gp120 of HIV presenting α-gal epitopes [64]. 

### 9.4. Immune Parameters as Limitations

The α-gal micelle treatment is subjected to immune-associated limitations, the same as any other cancer immunotherapy. The main general limitations are the immunogenic potency of the various TAs and the suppressive effects of chemotherapy treatments on the immune system. The immunogenicity of TAs widely varies in cancer patients with the same type of cancer. Evidently, the success of immunotherapy treatments in patients is proportional to TA immunogenicity in the individual patient. Nevertheless, the α-gal micelle treatment is likely to increase low immunogenicity of TAs due to the increased amounts of TA molecules internalized into APCs, as demonstrated in this review. A second important limitation of the proposed treatment, the same as that of other immunotherapies, is the ability of the immune system to function properly and perform the various processes, described in Figure 1A, in patients that received chemotherapy or irradiation therapy shortly prior to the α-gal micelle treatment. Therapies which have immunosuppressive effects may decrease or diminish the efficacy of the α-gal micelle therapy if delivered within a short period prior to this treatment. A useful measure for the efficacy of the immune system prior to the treatment may be ELISA, measuring titers of the natural anti-Gal antibody [100] in comparison to that in a serum sample obtained before the initiation of the chemotherapy or irradiation protocols. A decrease in the titer following chemotherapy or irradiation will indicate the immunosuppression of the immune system at the time of analysis. A significant decrease in anti-Gal activity will suggest impaired functions of the immune system.

### 9.5. Limitations Due to α-gal Syndrome

Many of the individuals with an allergy to meat (beef, pork and lamb) were found to produce IgE anti-Gal antibodies following bites by the tick *Ambliyomma americanum* (also called the “lone star” tick) [101,102]. This allergy is induced as a result of the interaction between anti-Gal IgE and α-gal epitopes on glycolipids and glycoproteins released from the digested meat. Such an allergic response is called the “α-gal syndrome”. Thus, cancer patients who are to be treated with α-gal micelles should be questioned prior to treatment whether they have the α-gal syndrome or have an allergy to meat. Confirming this allergic response by a skin test with α-gal micelles [96] would indicate that the patient requires suppression of an allergic reaction prior to treatment with α-gal micelles. 

## 10. Conclusions

The overall objective of the α-gal micelle immunotherapy is to induce a protective immune response against metastatic cells by activating the cellular and humoral immune system against the full range of self-TAs, without the need for identifying these antigens. Pre-clinical studies with natural or synthetic α-gal micelles injected intra-tumoral in mice producing the anti-Gal antibody demonstrated the conversion of B16 melanoma TA into autologous vaccine. The injected micelles insert into the tumor cells within the treated melanoma lesion. The resulting expression of α-gal epitopes on the tumor cells is followed by anti-Gal binding to them, the activation of the complement system, and the recruitment of APCs by complement cleavage chemotactic peptides. Anti-Gal opsonized tumor cells are further targeted for robust uptake by the recruited APCs. The internalized TAs are processed within the APCs, which transport the MHC-presenting TA peptides to regional lymph nodes. This was found to result in the induction of effective anti-TA immune responses, which destroy distant metastases. Subsequent clinical trials with natural and synthetic α-gal micelles in cancer patients with advanced disease indicated that these micelles are not toxic, and their use is safe. However, the ability of this immunotherapy to induce a protective immune response against self-TAs in the patients was found to be lower than that observed in the pre-clinical animal model. It is probable that the efficacy of this immunotherapy greatly decreases in advanced stages of the disease when the total tumor mass in the primary tumor and in metastases is high. All patients approved for participation in the Phase 1 studies described above were in advanced stages of the disease and failed standard therapies. The studies in mice suggest that when the tumor mass is large, even an effective immune response cannot “catch up” with the expanding tumor metastases and destroy them. However, in earlier stages of the disease, the treatment in mice succeeded in destroying metastatic tumor cells distant from the treated lesion. 

The abscopal effects of α-gal micelle treatment in the experimental model suggest that an intra-tumoral injection of such natural or synthetic micelles may help in the prevention of disease progression due to eliciting an anti-TA protective immune response. The injection of α-gal micelles into the primary lesion may be performed upon its detection and few weeks prior to its resection (i.e., as a neoadjuvant therapy), or into several of the accessible metastases that appear after the resection of the primary tumor. The ability of this immunotherapy to increase the number of primed TA-specific T cells also synergizes with checkpoint-inhibitor therapies if delivered prior to the latter immunotherapies. As with many other immunotherapies, the suppression of the immune system by chemotherapy or irradiation may prevent full beneficial effects of the α-gal micelle immunotherapy.

## Figures and Tables

**Figure 2 pharmaceutics-16-01263-f002:**
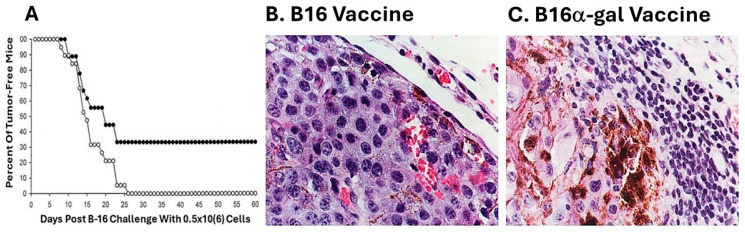
Protective anti-TA vaccination in GT-KO mice with B16_α-gal_ cells. (**A**) Mice free of melanoma lesions in the group immunized with 2 × 10^6^ B16_α-gal_ cells (●) or with B16 cells (○) and challenged 2 weeks later with 0.5 × 10^6^ live B16 cells (n = 27 mice per group). (**B**) Representative tumor lesion (~10 mm diameter) from a mouse immunized with irradiated B16 cells, 2 weeks prior to the challenge with live B16 cells, as (○) in **A**. (**C**) The same as in **B**, but the mouse was immunized with B16_α-gal_ cells 2 weeks prior to challenge, as the (●) mice, which developed a tumor in **A**. Note the multiple mononuclear cells surrounding the tumor cells, the vacuoles, and the melanin granules in the tumor cells adjacent to the infiltrating mononuclear cells. (H&E ×400) Modified with permission from ref. [69] 1999, Cancer Research AACR.

**Figure 4 pharmaceutics-16-01263-f004:**
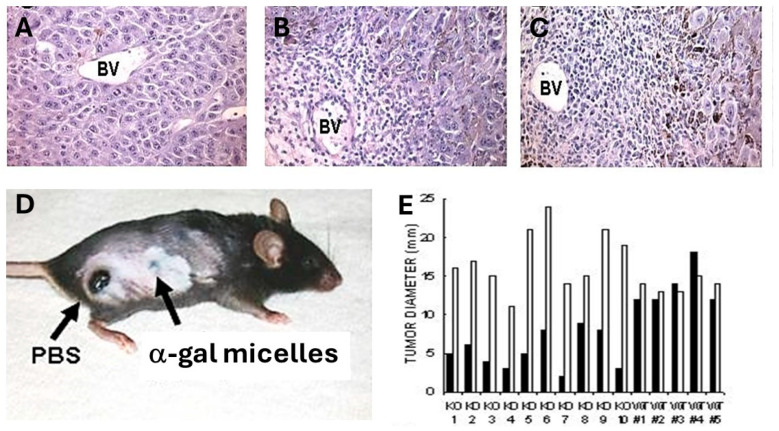
Effects of 1.0 mg α-gal micelle injection into representative B16 tumor lesions (~5mm in diameter) in anti-Gal-producing GT-KO mice (KO mice) or wild-type mice (WT mice). (**A**) The injection of PBS into control lesions results in no infiltration of mononuclear cells, as observed in B16 lesions, 7 days post injection. (**B**) Day 2 post injection of α-gal micelles into lesions of anti-Gal-producing GT-KO mice. Note the multiple mononuclear cells extravasating the blood vessel (BV). (**C**) Day 7 post injection of α-gal micelles. Note the marked increase in the number of extravasating mononuclear cells. (**D**) Two melanoma lesions on a representative GT-KO mouse, 10 days after the intra-tumoral injection of PBS (left lesion) or of α-gal micelles (right lesion). Note the near-complete regression of the right lesion vs. the increased size of the left lesion to ~12 mm. (**E**) A comparison of paired B16 lesions’ diameter in 10 GT-KO mice or 5 WT mice as in **D**, injected with α-gal micelles or with PBS, on day 10 post injection. Note the prevention of growth or decreased size in the lesions treated with α-gal micelles vs. the continuing growth of lesions injected with PBS in KO mice. In contrast, no differences in lesion growth post injection were observed in WT mice lacking the anti-Gal antibody. (**A**–**C**) H&E × 100 reproduced with permission from ref. [89] 2021 MDPI, and (**D**,**E**) were adapted with permission from ref. [87] 2018, Elsevier.

**Figure 5 pharmaceutics-16-01263-f005:**
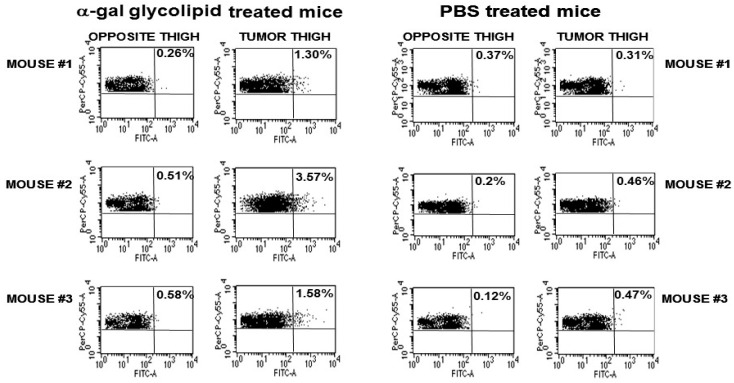
The APC-mediated transport of OVA as a surrogate TA from α-gal micelle-treated B16/OVA lesions to inguinal lymph nodes in anti-Gal-producing GT-KO mice. The lymph nodes were excised 2 weeks post treatment and APCs presenting the OVA immunodominant SIINFEKL peptide were detected by flow cytometry following the activation of *lac-Z* transgene under the IL2 promoter in activated CD8^+^ B3Z hybridoma T cells. The percentage of activated B3Z cells was determined by double staining CD8+ cells (PerCP red) and di-galactoside hydrolyzed by β-galactosidase (FITC green). Reproduced with permission from ref. [87] 2018, Elsevier.

**Figure 6 pharmaceutics-16-01263-f006:**
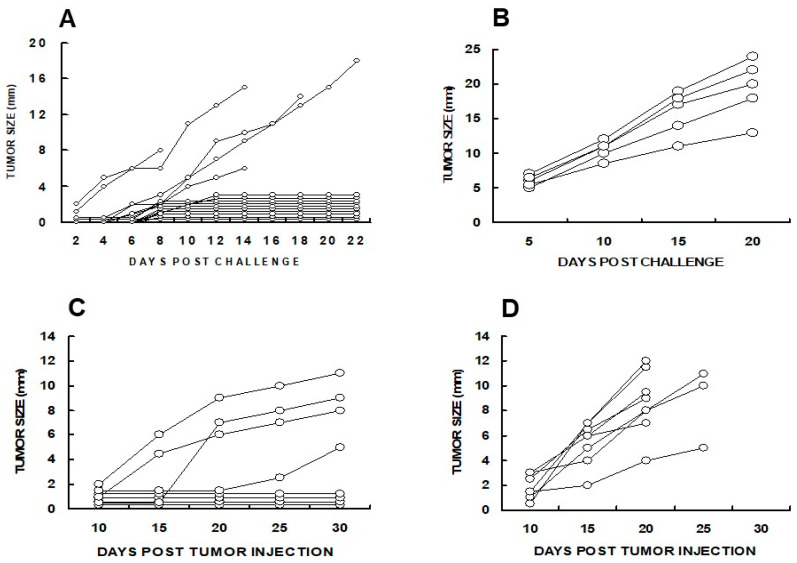
The prevention of distant lesions’ development by treatment of primary tumor lesions with α-gal micelles in anti-Gal-producing GT-KO mice. (**A**) Mice received three weekly intra-tumoral injections of 1.0 mg α-gal micelles into primary B16 melanoma lesions in the right flank. Subsequently, the mice were challenged with 0.5 × 10^6^ B16 cells in the left flank, simulating distant metastatic cells. The tumor growth of the challenging B16 cells was monitored. Ten of the fifteen mice injected with α-gal micelles displayed no development of a distant metastasis. (**B**) Control mice with primary tumor injected with PBS. All mice displayed metastatic tumor growth. (**C**) One million B16 melanoma cells were administered subcutaneously into the right flank as a “primary” tumor. At the same time, 1 × 10^4^ cells were administered to the left flank, simulating a “distant” micro-metastasis. The primary tumor was injected with α-gal micelles, as in **A**, and the injection was repeated a week later. Monitoring the development of a distant metastasis in the left flank indicated that the treatment prevented metastases’ growth in 4 of the 8 mice tested. (**D**) Tumor challenges as in (**C**), however, the right flank tumor received two PBS injections (n = 8 mice/group). All mice developed metastases. Adapted with permission from ref. [94] 2009, Springer Nature.

**Figure 7 pharmaceutics-16-01263-f007:**
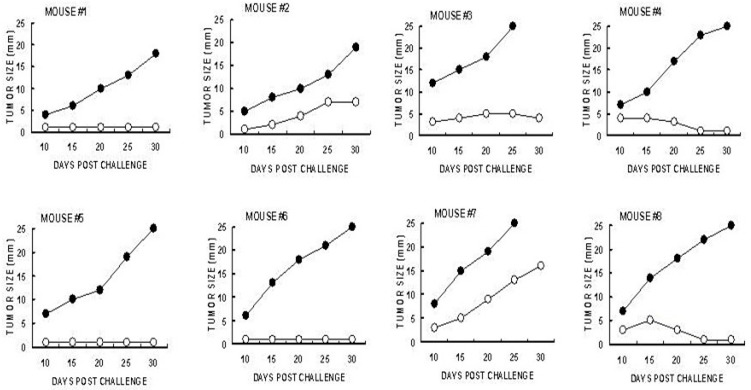
Protection against B16 tumor growth following α-gal micelle treatment is mediated by CD8+ T cells. Forty million splenocytes were transferred from anti-Gal GT-KO mice with B16 lesions treated with α-gal micelles, to naïve GT-KO recipient mice. These recipients were challenged subcutaneously with 0.5 × 10^6^ live B16 cells, 24 h prior to receiving the splenocytes. Tumor growth was monitored for 30 days. (○)—Transfer of total splenocytes, (●)—Transfer of splenocytes depleted of CD8+ T cells. The size of the growing tumors is described at various time points post adoptive transfer. Reprinted with permission from ref. [94] 2009, Springer Nature.

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
