# Peer review of "Self-Tumor Antigens in Solid Tumors Turned into Vaccines by α-gal Micelle Immunotherapy"

_pharmaceutics, 2024, doi:10.3390/pharmaceutics16101263_

Round 1

Reviewer 1 Report

Comments and Suggestions for Authors

The author has designed this paper to discuss the  possibilty of using the Self-Tumor Antigens in Solid Tumors Turned Into Vaccines by alpha-gal Micelles Immunotherapy, the method showed promising outcomes. However, there are a number of minor concerns with the manuscript that need to be addressed.

1.     The private antigens or publicly shared antigens can be referred to as TAA (Tumor-Associated Antigens) or TSA (Tumor-Specific Antigens).

2.     There is no reference provided for the information in lines 41-48.

3.     It would be beneficial if the author could include the mechanism of activation for APCs and other immune cells, or perhaps provide an image to illustrate this process which Is the most important part of cancer vaccine mechanism.

4.     In the introduction author can add the importance of α-Gal Micelles in immunotherapy by highlighting the role of α-gal epitopes and the natural anti-Gal antibody in the concept of α-gal micelles as a promising approach for enhancing the immune system's ability to target and destroy tumor cells. 

5.     The author mentiobed in the line 150, These recruited APC effectively phagocytose tumour cells and cell membranes opsonized by anti-Gal IgG following binding of the Fc tail of this antibody to Fcγreceptors (FcγR) on APC and binding of C3b on the opsonized cells to C3b receptors (CR1) on APCthe APC engulfment with FcγR, but they did not indicated which receptor because FcγRI is involved in the phagocytosis of opsonized cells (such as the tumor cells in this case) and in the activation of immune responses. FcγRIIb is an inhibitory receptor, which helps to regulate and limit immune responses, preventing excessive inflammation. FcγRIIIa is important for antibody-dependent cellular cytotoxicity (ADCC), a process by which NK cells kill opsonized target cells, and it also contributes to phagocytosis by macrophages.

6.     It would be better if the author could increase the quality of the images, for example fig 1 A.

7.     In the line 364 author mentiobed that,  Production of alpha-gal micelles, but there is the lack of characterization data (e.g., size, purity, composition, number of α-gal epitopes per micelle) leaves open questions about the consistency and quality of the α-gal micelles produced. Techniques such as mass spectrometry, NMR, or HPLC)could be used to verify the purity and composition of the α-gal glycolipids. Additionally, DLS or electron microscopy could provide information on the size and uniformity of the micelles.

8.     In section 5-5, while the experiment shows increased activation of CD8+ T cells, it does not explore the underlying mechanisms by which α-gal micelles enhance the uptake and processing of TA by APCs. The experiment shows that α-gal micelles enhance the uptake and processing of OVA, but it does not explore the underlying mechanisms by which this enhancement occurs.

9.     It seems author need to add the challenges and considerations scetion in this paper to address any remaining challenges, such as the potential variability in patient responses or the need for further optimization of treatment protocols, and discuss the potential limitations of the current research and areas where more data is needed.

Author Response

I thank the reviewer for the comments and suggestions which improve the manuscript. The red font indicates added text to the revision

Comments and Suggestions for Authors

The author has designed this paper to discuss the  possibility of using the Self-Tumor Antigens in Solid Tumors Turned Into Vaccines by alpha-gal Micelles Immunotherapy, the method showed promising outcomes. However, there is a number of minor concerns with the manuscript that need to be addressed.

  1. The private antigens or publicly shared antigens can be referred to as TAA (Tumor-Associated Antigens) or TSA (Tumor-Specific Antigens).

Response: Because of the controversies regarding the type of tumor antigens included under the TAA and TSA terms, the words “private” and “public” tumor antigens were deleted and only the term “tumor antigens” i.e., “TA” has been used in the revised manuscript.

  1. There is no reference provided for the information in lines 41-48.

 Response: References 1-5 were added regarding the information in lines 41-45. The sentence  in lines 46-48 “In a proportion of patients with solid tumors theimmune system “succeeds” by itself to mount a protective immune response against the patient’s TA.” has been deleted from the revised manuscript because of insufficient clear support to this statement.

  1. It would be beneficial if the author could include the mechanism of activation for APCs and other immune cells, or perhaps provide an image to illustrate this process which Is the most important part of cancer vaccine mechanism.

Response: As added in lines 166 to 171 and detailed in the added references 56 and 57  “The Fcg receptors (FcgR) on the recruited APC bind Fc tails of the opsonizing anti-Gal. This Fc/FcgR interaction induces rearrangements in the actin cytoskeleton of the APC that lead to the phagcytosis of the opsonized tumor cells and cell membranes [56]. Binding of C3b on the opsonized cells to C3b receptors (CR1) on APC induces a similar actin cytoskeleton and phagocytosis of the cells bearing C3b by the APC [57].” In addition, the specific steps of the mechanism induced by alpha-gal micelles treatment of tumor were detailed in Figure 1A and in the corresponding text in lines  153 to 177 of the revised manuscript. The division of the process into Steps 1-4 was introduced in the revised manuscript.

  1. In the introduction author can add the importance of α-Gal Micelles in immunotherapy by highlighting the role of α-gal epitopes and the natural anti-Gal antibody in the concept of α-gal micelles as a promising approach for enhancing the immune system's ability to target and destroy tumor cells. 

Response: As suggested by the reviewer, the importance of alpha-gal micelles was added at the end of the Introduction in lines   95 to 103  as follows: “As detailed in this review, the main advantages of the vaccination immunotherapy with a-gal micelles are: 1. The simplicity  of the treatment which involves only intra-tumoral injection of a-gal micelles and does not require any in vitro culturing of dendritic cells and pulsing these cells with TA or fusing them with tumor cells, 2. The possible use of this vaccine as both neoadjuvant immunotherapy and for inducing a protective immune response against metastatic cells, 3. The ability to elicit an immune response against the full range of TA of each patient (i.e., a personalized vaccine) without the need to identify any of the TA, and 4. The ability of this therapy to synergize with checkpoint inhibitors therapies.” In addition, The role of α-gal epitopes and the natural anti-Gal antibody in the concept of α-gal micelles was defined in the 4 step processes detailed in lines 153 to 177.

  1. The author mentioned in the line 150, These recruited APC effectively phagocytose tumour cells and cell membranes opsonized by anti-Gal IgG following binding of the Fc tail of this antibody to Fcγreceptors (FcγR) on APC and binding of C3b on the opsonized cells to C3b receptors (CR1) on APCthe APC engulfment with FcγR, but they did not indicated which receptor because FcγRI is involved in the phagocytosis of opsonized cells (such as the tumor cells in this case) and in the activation of immune responses. FcγRIIb is an inhibitory receptor, which helps to regulate and limit immune responses, preventing excessive inflammation. FcγRIIIa is important for antibody-dependent cellular cytotoxicity (ADCC), a process by which NK cells kill opsonized target cells, and it also contributes to phagocytosis by macrophages.

Response: As indicated in lines 199 to 200 “This phagocytosis was found to be mediated by the FcgammaR1 on the human APC [66]”.

  1. It would be better if the author could increase the quality of the images, for example fig 1 A.

 Response: The figures were prepared with the highest resolution of the computer. As for the difficulty in reading the structure of the glycolipid molecule in Figure 1A, the following sentence was added in line 215 of the legend for this figure “The detailed structure of the glycolipid with the alpha-gal epitope is presented in Figure 3A at a larger magnification”.

  1. In the line 364 author mentioned that,  Production of alpha-gal micelles, but there is the lack of characterization data (e.g., size, purity, composition, number of α-gal epitopes per micelle) leaves open questions about the consistency and quality of the α-gal micelles produced. Techniques such as mass spectrometry, NMR, or HPLC)could be used to verify the purity and composition of the α-gal glycolipids. Additionally, DLS or electron microscopy could provide information on the size and uniformity of the micelles.

 Response: The production of alpha-gal micelles was detailed in the revised  lines 367 to 378 and is detailed in in ref. [55]. As indicated in that section, the analysis of the glycolipids in the micelles was performed by thin layer chromatography (TLC) and immunostaining of the TLC plates with mouse and human anti-Gal. We could not provide clear electron microscopy information on the structure of the micelles because of their very small size ranging from 2 nm (20 A) to 20 nm (200 A).

  1. In section 5-5, while the experiment shows increased activation of CD8+ T cells, it does not explore the underlying mechanisms by which α-gal micelles enhance the uptake and processing of TA by APCs. The experiment shows that α-gal micelles enhance the uptake and processing of OVA, but it does not explore the underlying mechanisms by which this enhancement occurs.

 Response: The underlying mechanisms were detailed in the text in lines 459 to 472 of the revised manuscript..

  1. It seems author need to add the challenges and considerations section in this paper to address any remaining challenges, such as the potential variability in patient responses or the need for further optimization of treatment protocols, and discuss the potential limitations of the current research and areas where more data is needed.

Response: In accordance with this suggestion of the reviewer, a new section (Section 9) entitled “Advantages and  limitations” was added in lines 695 to 785. Due to this revision, the Conclusions section was shortened to prevent redundancies.

Reviewer 2 Report

Comments and Suggestions for Authors

The author has conducted research for a long time in the field of improving the efficacy of cancer immunotherapy by transporting alpha-gal to improve immune function, even though alpha-gal is not expressed in the human body, and has published a number of academic papers. Although the increase in immunity using the introduction of alpha-gal stopped in phase 1 clinical trials, I believe that the paper is worthy of publication as it summarizes important research. In order to improve the paper, I would like to reinforce the following points.

1. In the review, the author only mentioned the increase in immunity in anti-cancer research using α-gal, but it would be nice if he explained it with examples of other representative agents (vaccines) that can increase immunoreactivity against cancer cells.

2. (Essential) The author used 7 figures in this review paper. These figures have already been used as figures in papers published by the author. It is said that the permission has been obtained. Is there any permission letter from the academic journal? Author must submit the permission letter.

3. Why do tumors that do not form tumor masses do not progress beyond clinical phase II? The author needs to comment on this in more detail. Why are there no clinical trials related to alpha-gal therapy in blood cancer (leukemia, lymphoma)?

4. What is the major limitation of alpha-gal mediated immunotherapy? Author need put the limitations of it in this manuscript.

5. Author need carefully check typos in Manuscript.  

Line 368 antennae, ?

Author used mixed hr and h.

Line 513 Adapated.

Line 567 duplicated “and”

Line 648 1-mg “-“ should be removed.

Line 663 performd

Line 712 syndrom

Line 721 effectve

Author needs to enhance image quality of Fig 3A, B

Redundant description 1

- Line 21 APC (dendritic cells and macrophages)

- Line 63 antigen presenting cells (APC) such as dendritic cells and macrophages.

-Line 198 APC (dendritic cells and macrophages).

Redundant description 2

There are several similar descriptions regarding APC transport to (regional) lymph-nodes.

Author Response

I thank the reviewer for the comments and suggestions which improve the manuscript. The red font indicates added text to the revision.

The author has conducted research for a long time in the field of improving the efficacy of cancer immunotherapy by transporting alpha-gal to improve immune function, even though alpha-gal is not expressed in the human body,  and has published a number of academic papers. Although the increase in immunity using the introduction of alpha-gal stopped in phase 1 clinical trials, I believe that the paper is worthy of publication as it summarizes important research. In order to improve the paper, I would like to reinforce the following points.

  1. In the review, the author only mentioned the increase in immunity in anti-cancer research using α-gal, but it would be nice if he explained it with examples of other representative agents (vaccines) that can increase immunoreactivity against cancer cells.

Response: The comparison to other vaccines is included in the revised lines 95 to 103 “As detailed in this review the main advantages of this vaccination immunotherapy is: 1. The simplicity of the treatment which involves only intratumoral injection of a-gal micelles and does not require any in vitro culturing of dendritic cells and pulsing these cells with TA, 2. The possible use of this vaccine as both neo-adjuvant immunotherapy and for inducing a protective immune response against metastatic cells, 3. The ability to elicit an immune response against the full range of TA of each patient (i.e., a personalized vaccine) without the need to identify any of the TA, and 4. The ability of this therapy to synergize with checkpoint inhibitors therapies”.

  1. (Essential) The author used 7 figures in this review paper. These figures have already been used as figures in papers published by the author. It is said that the permission has been obtained. Is there any permission letter from the academic journal? Author must submit the permission letter.

Response: The author has obtained permissions for use of all previously published figures and will provide these permission letters to the editorial office upon request, if the manuscript is accepted for publication.

  1. Why do tumors that do not form tumor masses do not progress beyond clinical phase II? The author needs to comment on this in more detail. Why are there no clinical trials related to alpha-gal therapy in blood cancer (leukemia, lymphoma)?

Response: As a PhD researcher in a clinical system, I have not succeeded in convincing hematologists to perform a clinical trial with this immunotherapy evaluation in leukemia and lymphoma, despite the IND approval I received from the FDA.

  1. What is the major limitation of alpha-gal mediated immunotherapy? Author need put the limitations of it in this manuscript.

Response: In accordance with the reviewer’s request (as well as that of Reviewer #1), the   limitations have been included in the new section added to the revised manuscript- Section 9 entitled “Advantages and limitations” in lines 695 to 785.

  1. Author need carefully check typos in Manuscript.  

Line 368 antennae, ?

Author used mixed hr and h.

Line 513 Adapated.

Line 567 duplicated “and”

Line 648 1-mg “-“ should be removed.

Line 663 performd

Line 712 syndrom.

Line 721 effectve

Author needs to enhance image quality of Fig 3A, B

Redundant description 1

- Line 21 APC (dendritic cells and macrophages)

- Line 63 antigen presenting cells (APC) such as dendritic cells and macrophages.

-Line 198 APC (dendritic cells and macrophages).

Redundant description 2

There are several similar descriptions regarding APC transport to (regional) lymph-nodes.

Response: I thank the reviewer for pointing out the typos which were corrected in the revised manuscript. The term “antennae” is the plural of  “antenna” The redundant descriptions of APC were deleted. The images in this manuscript were included at the highest resolution of my computer.

Reviewer 3 Report

Comments and Suggestions for Authors

1. The review heavily summarizes existing studies without providing sufficient critical evaluation of the strengths, limitations, or controversies in the field. There is a lack of discussion on conflicting results or alternative interpretations of key findings.

2. The manuscript dedicates excessive space to describing the mechanism of anti-Gal/α-gal micelles, while other important aspects of cancer immunotherapy, such as recent advancements in checkpoint inhibitors or CAR-T cell therapy, are only briefly mentioned or overlooked.

3. The review provides a thorough summary of the mechanism behind α-gal micelles, but could you incorporate a more critical analysis of the existing literature? For example, are there any significant limitations, controversies, or areas where the evidence is conflicting that should be highlighted?

4. Although the review focuses on the use of α-gal micelles, it does not adequately compare this approach with other emerging cancer vaccine strategies. A broader discussion of how α-gal micelles fit within the larger landscape of immunotherapies would be beneficial. How does the α-gal micelle approach compare to other cancer vaccine strategies, such as neoantigen vaccines or dendritic cell-based therapies? Could you discuss the relative advantages and disadvantages of α-gal micelles within the broader context of cancer immunotherapy?

5. The review focuses primarily on the α-gal micelles approach. Are there other complementary or competing strategies in the field of cancer immunotherapy that warrant more attention in this review? How might these other strategies interact with or enhance the efficacy of α-gal micelles?

6. The review does not adequately address future research directions or potential clinical applications. A more detailed discussion on the gaps in knowledge, challenges for translation to clinical practice, and suggestions for future studies would add significant value. The section on future directions is somewhat limited. Could you expand this to discuss specific research questions that remain unanswered? What are the next steps needed to move α-gal micelles from preclinical studies to broader clinical application?

7. The manuscript’s structure could be improved for better readability. Some sections are dense with technical details, while others are relatively superficial. Additionally, there are abrupt transitions between sections that disrupt the flow of the narrative.

Comments on the Quality of English Language

Moderate English language revision is required 

Author Response

I thank the reviewer for the comments and suggestions which improve the manuscript. The red font indicates added text to the revision.

  1. The review heavily summarizes existing studies without providing sufficient critical evaluation of the strengths, limitations, or controversies in the field. There is a lack of discussion on conflicting results or alternative interpretations of key findings.

Response: The comparison to other vaccines is included in the revised lines 95 to 103 “As detailed in this review the main advantages of this vaccination immunotherapy is: 1. The simplicity of the treatment which involves only intra-tumoral injection of a-gal micelles and does not require any in vitro culturing of dendritic cells and pulsing these cells with TA, 2. The possible use of this vaccine as both neo-adjuvant immunotherapy and for inducing a protective immune response against metastatic cells, 3. The ability to elicit an immune response against the full range of TA of each patient (i.e., a personalized vaccine) without the need to identify any of the TA, and 4. The ability of this therapy to synergize with checkpoint inhibitors therapies”.

The limitations have been included in the new section added to the revised manuscript- Section 9 entitled “Advantages and limitations” in lines 695 to 785.

  1. The manuscript dedicates excessive space to describing the mechanism of anti-Gal/α-gal micelles, while other important aspects of cancer immunotherapy, such as recent advancements in checkpoint inhibitors or CAR-T cell therapy, are only briefly mentioned or overlooked.

Response: This review aims to describe the alpha-gal micelles immunotherapy which is a novel one and the clinical trials with it are so far very limited and described in the review.  Thus, other than the advantages of and limitations described in section 9 of the revised manuscript there is no additional information that enables comparison with other cancer immunotherapy methods. The other therapies are mentioned in quoted references [11, 100], and in the sections describing experimental studies on synergism with checkpoint inhibitors therapies (lines 721 to 738). The space constraints for this review do not allow for a detailed  description of other therapies. CAR-T cell therapy is not mentioned in since it is not considered as a vaccine method.

  1. The review provides a thorough summary of the mechanism behind α-gal micelles, but could you incorporate a more critical analysis of the existing literature? For example, are there any significant limitations, controversies, or areas where the evidence is conflicting that should be highlighted?

Response: The alpha-gal micelles immunotherapy was developed in my lab and the only other independent study which repeated our studies using the mouse experimental model and synthetic alpha-gal micelles is the study described in ref. 43 and in section 7 of the review. The only clinical trials performed so far with alpha-gal micelles are those described in sections 6 and 8.

  1. Although the review focuses on the use of α-gal micelles, it does not adequately compare this approach with other emerging cancer vaccine strategies. A broader discussion of how α-gal micelles fit within the larger landscape of immunotherapies would be beneficial. How does the α-gal micelle approach compare to other cancer vaccine strategies, such as neoantigen vaccines or dendritic cell-based therapies? Could you discuss the relative advantages and disadvantages of α-gal micelles within the broader context of cancer immunotherapy?

Response: As indicated in response #3 above, the only clinical trials that have been performed are described in sections 6 and 8. The results obtained so far are insufficient for comparison with other therapies used in cancer patients. As for the advantages and limitations of the alpha-gal micelles therapy, they are described in line 95 to 103 of the Introduction and in the added section 9 of the manuscript. The neoantigen therapies are mentioned in lines 74 to 80 and the corresponding references 16-19, and line 698.. Reference to dendritic cell therapy is included in revised lines 96 to 98.

  1. The review focuses primarily on the α-gal micelles approach. Are there other complementary or competing strategies in the field of cancer immunotherapy that warrant more attention in this review? How might these other strategies interact with or enhance the efficacy of α-gal micelles?

Response: I do not know of other treatments which opsonize tumor cells within tumor lesions by a natural antibody and target tumor cells and cell membranes for robust uptake by the recruited APC. As for complementary in the field of cancer immunotherapy, as described in section 9.2. of the revised manuscript, alpha-gal micelles treatment can amplify the efficacy of checkpoint inhibitors treatment (lines 721 to 738).  

  1. The review does not adequately address future research directions or potential clinical applications. A more detailed discussion on the gaps in knowledge, challenges for translation to clinical practice, and suggestions for future studies would add significant value. The section on future directions is somewhat limited. Could you expand this to discuss specific research questions that remain unanswered? What are the next steps needed to move α-gal micelles from preclinical studies to broader clinical application?

Response: As suggested in the review (lines 715 to 719 and 734 to 738), the future steps should be performing Phase II clinical trials. The information in section 9 that was added to the revised manuscript directs future researchers/clinicians in studying this immunotherapy in cancer patients who may most benefit from it and performing combined therapies with checkpoint inhibitors treatment.

  1. The manuscript’s structure could be improved for better readability. Some sections are dense with technical details, while others are relatively superficial. Additionally, there are abrupt transitions between sections that disrupt the flow of the narrative.

Response: Some connecting sentences were added to improve the flow of the narrative in lines 124, 153, 473, and 652.

Round 2

Reviewer 3 Report

Comments and Suggestions for Authors

The authors have covered all the raised comments. 

Comments on the Quality of English Language

Minor English language revision is required